# The Association between Video Game Type and Aggressive Behaviors in Saudi Youth: A Pilot Study

**DOI:** 10.3390/bs12080289

**Published:** 2022-08-15

**Authors:** Majid A. Aleissa, Shuliweeh Alenezi, Hassan N. Saleheen, Sumayyah R. Bin Talib, Altaf H. Khan, Shatha A. Altassan, Ahmed S. Alyahya

**Affiliations:** 1National Family Safety Program, King Abdulaziz Medical City, Ministry of National Guard Health Affairs, Riyadh 11426, Saudi Arabia; 2Department of Pediatrics, King Abdullah Specialized Children’s Hospital, Riyadh 11426, Saudi Arabia; 3King Abdullah International Medical Research Center, Riyadh 11481, Saudi Arabia; 4Department of Psychiatry, College of Medicine, King Saud University, Riyadh 11451, Saudi Arabia; 5SABIC Psychological Health Research and Applications Chair (SPHRAC), College of Medicine, King Saud University, Riyadh 12372, Saudi Arabia; 6Department of Psychiatry, Eradah Complex for Mental Health, Riyadh 12571, Saudi Arabia

**Keywords:** adolescence, aggression, video gaming, Saudi Arabia

## Abstract

Video gaming is a popular source of entertainment among children and adolescents. Although the Middle East is home to one of the fastest growing communities of video game users, most of the research established on this topic has been carried out through small scale studies. Our aim in this study is to assess the prevalence of video game use and its association with aggressive behaviors among adolescents in Saudi Arabia. This is a cross-sectional study involving boys and girls (aged 15–18 years) in both private and public secondary high schools in Riyadh, Saudi Arabia. Each participant completed a self-administered modified version of the aggression questionnaire, which consisted of 29 items scored on a 5-point Likert scale. This questionnaire assessed aggressive behaviors domains: physical aggression, anger, hostility, and verbal aggression and types of videogames and time of use. A total of 485 students were included in this study. The mean age of participants was 16.5 ± 0.9 years; 48% were boys. Adolescents who participated in action games had higher mean verbal (*p* < 0.01) and physical aggression (*p* < 0.01) scores. Adventure game players had significantly higher mean scores in all four types of aggressive behavior (*p* < 0.01). Participants who played simulation games had higher mean verbal aggressiveness (*p* < 0.01). Adolescents who participated in sports games had greater mean levels of anger (*p* = 0.01) and physical aggression (*p* = 0.01). Those who played strategy/puzzle games reported significantly higher mean scores of anger (*p* < 0.01), hostility (*p* = 0.01), and verbal aggression (*p* = 0.01). Females were more likely to show higher mean anger (*p* < 0.01) scores, whereas males were more likely to show higher mean physical aggression scores (*p* < 0.01). Conclusions: Our results do suggest that playing video games increases adolescent aggressive behaviors, which has been supported by other studies. We recommend educating parents on the pros and cons of playing video games and that parents schedule and limit the time their children spend playing video games.

## 1. Introduction

Video gaming is a popular source of entertainment among children and adolescents [1]. In the United States, children and adolescents play video games an average of 7 h per week, with an addiction prevalence of approximately 6%. Human and fantasy violence games account for approximately 50% of children’s favorite games, with sports violence contributing another 16–20% for boys and 6–15% for girls [2,3,4]. Even though males spend more time playing violent video games than females, video games can increase aggressiveness in both sexes. Moreover, the longer that children are exposed to violent video games, the more likely they are to adopt aggressive behaviors and thoughts [5]. In terms of age, a similar study conducted on adults showed that being male and in the younger age group (aged 16–21) increases the likelihood of problematic video game-related behavior [6]. The Middle East is home to one of the fastest growing communities of video game users. A study conducted in Iran in 2010 showed that 47% of children have played one or more intensely violent video games [1].

The American Psychological Association (APA) considered violent video games to be a predictive factor for aggressive behavior [7]. There are many psychological theories (e.g., social learning theory) that have been established to better understand the influence of violent video games on aggression. One model, the General Aggression Model (GAM), was specifically formulated in order to understand this association. This model suggests that the relationship between exposure to a situational variable (e.g., violent media) and aggression is mediated by cognition [8]. Consistently, it was showed that exposure to violent video games is positively associated with using moral disengagement mechanisms, which, in turn, can encourage individuals to cognitively reconstruct aggression. This occurs by making the outcome of their actions appear less harmful, minimizing their role in the outcome, and by reducing their recognition for victims [9]. Similarly, another study showed that youth aggression is positively related to exposure to violent video games, which is mediated by normative beliefs about aggression [10]. There are a few studies that have differentiated between violent and non-violent video games, stating that violent video games can desensitize people to seeing aggressive behavior and decrease prosocial behaviors (e.g., empathy) [5]. This association was also true for both aggressive cognition (implicit) and aggressive behavior (explicit), even if the adolescent was briefly exposed to a violent video game [11]. Other associations are also apparent between video gaming and elevated drug and alcohol use, lower interpersonal relationship quality, poor performance at school, reduced sleep time, and suicidal ideation [12]. Nonetheless, the relationship between violent video games and aggressive conduct in adolescents seem to be still up for discussion [13].

The types of video games played in relation to aggressive behavior were examined in the literature with a noticeable interest in violent games impact. Most of the evidence point to a relationship between violent video games and aggression in youth. For example, a meta-analysis of 24 studies with over 17,000 participants, showed that violent video games were associated with aggression [14]. However, a recent study that included different types of video games showed that violent and competitive video games were favorably associated with self-reported physical aggression. In contrast, simulation games were found to be negatively associated. This study also found that physical aggression was not associated with video game genres, including strategy, sports, offline shooter, racing, adventure, puzzle, and platform games [15]. These findings back up the theory that having both competition and violence in a game enhances the likelihood of physical aggression.

Most of the research established on video gaming in the Middle East has been carried out through small-scale studies (i.e., non-longitudinal scales) and didn’t assess the level of aggression or association with different types of video games. Instead, generalized questionnaires were used to assess social health, quality of life, or behavior and attitude [16,17,18]. There is a lack of national data or documentation to prove the impact of video games on mental health and aggressive behaviors. Therefore, the aim of this study is to assess the prevalence of video game use and its association with aggressive behaviors among adolescents in Saudi Arabia. Although this is also a non-longitudinal study, it assesses the relationship between playing different types of video games and aggression scores using a specifically designed survey.

## 2. Materials and Method

### 2.1. Design, Participants, and Procedure

A cross-sectional study was undertaken to identify the prevalence of video game use and its association with aggressive behaviors among adolescents in Riyadh, Saudi Arabia. Boys and girls aged 15–18 in both private and public secondary high schools were invited to participate in the study. A multistage stratified sampling technique was used. The city was stratified into five geographical areas: East, West, North, South, and Central. Each area was stratified into private and public schools, then stratified into middle and high schools, and finally into boys’ and girls’ schools. In the first phase, a cluster of middle and high schools from the public and the private boys’ and girls’ schools were randomly selected. A total of twenty schools were approached, four in each geographical area. However, four school’s principles refused to participate. In the second phase, classes were randomly selected from the schools selected in the first phase, and all students in these classes were delivered an anonymous questionnaire to complete at the schools. It took 15–20 min for the participants to complete the questionnaire. Prior to administering the questionnaire, a standardized training was conducted for test administrators.

### 2.2. Measures

Each participant self-completed a modified version of the aggression questionnaire, which was originally developed by Buss and Perry (1992) and has been widely utilized and validated to investigate aggressiveness in various populations [19]. The modified version of the Buss and Perry questionnaire used in this study was in Arabic and some additional items were included: demographic data, types of video games played, and video game time exposure (see Table 1). The instrument consists of 29 items that are scored with a 5-point Likert scale, with 1 representing “extremely uncharacteristic of me” and 5 representing “extremely characteristic of me.” The questionnaire has four sub-scales:

*Physical aggression* assesses hurting others physically and consists of nine items (score range from 9 to 45);

*Anger* assesses the affective aspect of aggression and consists of seven items (score range from 7 to 35);

*Hostility* assesses the cognitive aspects of aggression and consists of eight items (score range from 8 to 40);

*Verbal aggression* assesses hurting others verbally and consists of five items (score range from 5 to 25).

Two additional open-ended questions were included to assess if all the items of the instrument were clear to the students. These questions were (a) *When answering the survey, were there some unclear or difficult questions for you?* and (b) *Do you have any comments or additional information to disclose?* The questionnaire was translated into Arabic and back-translated for comparison. The instrument was tested on a group of adolescents to ensure an understanding of the questionnaire and clarity of the answer choices. The translated version was amended as a result of this focus group. The overall reliability of the translated instrument was 0.848, which indicates that the instrument yielded consistent results since reliability is accepted (good).

This pilot cross-sectional study attempted to evaluate the usability and acceptability of the Arabic version of the instrument for a future national surveillance study. It also aims to identify ethical and methodological challenges to measure aggressive behaviors among Saudi adolescents. Since this is a descriptive study, no sample size was calculated. A feasibility sample of 500 adolescents has been selected so that our study will have enough power to be able to detect a prevalence of 50% with a 5% margin of error and 95% confidence level (according to the studies in other Muslim countries, the prevalence of video game use is reported to be 45.7% to 65.5% [10,11,12]).

### 2.3. Data Analysis

The data were analyzed using SPSS version 23 (IBM Corp. Released 2015. IBM SPSS Statistics for Windows, Version 23.0. Armonk, NY, USA: IBM Corp.). Participants were categorized by sociodemographic status, including age, sex, nationality, living arrangement, and caregivers. Scores for different types of aggressive behaviors were calculated by summing scores for corresponding items. Comparisons of scores according to age, sex, nationality, and types of video games used were performed via an independent samples *t*-test followed by Bonferroni correction for multiple comparisons (α = 0.02). Cohen’s *d* was calculated as a measure of the effect size. Correlations between number of video games played and self-reported aggression scores was performed by Pearson’s correlation test. Logistic regression analysis was performed with aggressive behaviors as the dependent variable (1 = Yes and 0 = No).

## 3. Results

We approached 577 students, of whom 502 completed the survey (87% participation rate). Out of 502 students, 17 were above the age limit, so we excluded them from the study. In the end, we analyzed the data of 485 students. Demographic information and hours spent playing video games per day are shown in Table 1.

Table 2 shows a comparison of aggression scores according to different types of video games played. Participants who played action games exhibited higher mean verbal (*p* < 0.01) and physical aggression (*p* < 0.01) scores relative to the scores observed for those who did not play action games and the effect size showed minor difference (Cohen’s *d* = 0.2). Those who played adventure games showed higher mean scores in all four types of aggressive behavior (*p* < 0.01) compared to those who did not play adventure games (Cohen’s *d* = 0.2–0.3). Higher mean verbal aggression (*p* < 0.01) was found among participants who played simulation games compared to those who did not play simulation games. The observed difference was of medium effect (Cohen’s *d* = 0.5). Participants playing sports games exhibited higher mean anger (*p* = 0.01) and physical aggression scores (*p* = 0.01) (Cohen’s *d* = 0.2–0.3). Those who played strategy/puzzle games reported higher mean scores of anger (*p* < 0.01), hostility (*p* = 0.01), and verbal aggression scores (*p* = 0.01) compared to those who did not play strategy/puzzle games with a small effect size (Cohen’s *d* = 0.2–0.3).

Females were more likely to play adventure and strategy/puzzle games (*p* < 0.01), while males were more likely to play sports and action games (*p* < 0.01). Regarding frequency of video game playing, males and Saudi students were more likely to play video games on a daily basis (Table 3).

A comparison of aggression scores according to demographic characteristics of the participants are also shown in Table 3. Females were more likely to show higher mean anger scores (*p* < 0.01) with medium effect size (Cohen’s *d* = 0.5) and males were more likely to show higher mean physical aggression scores (*p* < 0.01) with small effect size (Cohen’s *d* = 0.3). In terms of nationality, non-Saudi students reported higher mean verbal aggression scores relative to those reported by Saudi students (*p* < 0.01) and the effect size value showed a minor difference (Cohen’s *d* = 0.3). Females who played action and adventure games were more likely to show higher mean anger scores (*p* < 0.01) (Cohen’s *d* = 0.4–0.6), whereas males who played similar games were more likely to show higher mean physical aggression scores (*p* < 0.01) (Cohen’s *d* = 0.3–0.4).

A significant weak correlation in the positive direction (*r* = 0.11, *p* < 0.05) was found between total number of games played and anger scores. Furthermore, the results indicate that there was a significant weak correlation in the positive direction (*r* = 0.13, *p* < 0.01) between total number of games played and hostility score, total number of games played and verbal aggression score (*r* = 0.20, *p* < 0.01), total number of games played and physical aggression score (*r* = 0.16, *p* < 0.01) and total number of games played and overall aggression score (*r* = 0.18, *p* < 0.01).

Compared to participants with no history of playing video games and adjusting for age, gender, nationality, and duration of video games played, participants who reported playing action games were 3.5 times more likely to have physical aggression and those who reported playing adventure games were 3.1 times more likely to have anger (Table 4).

## 4. Discussion

This study is the first to examine the relationship between aggression and different types of video games in Saudi youth. We were also interested in assessing gender and sociodemographic variables in the context of aggression and video gaming. One of the main criticisms that has been leveled against video gaming is that most video games have aggressive elements, regardless of game type. This led to the belief that people become more aggressive after playing video games [20,21]. However, in our study, we found that physical aggression scores were higher in those who play action, sports, and strategy/puzzle games. Similarly, verbal aggression was reported to be higher in those who play action and strategy/puzzle games, but also in youth playing adventure and simulation games. The presence of an element of violent behavior in different genres of games may explain the similar aggression scores among users of different types of games in our study [22].

Furthermore, in our sample, aggression scores were higher among those who play video games vs. those who do not. A statistical significance was established when comparing those who play action games with participants with no history of playing video games. Participants who play action video games were 3.5 times more likely to have physical aggression. Moreover, those who play adventure games were 3.1 more likely to experience anger. This is supported by a recent meta-analysis which showed that playing violent video games is associated with an increase in the measures of serious aggressive behavior (i.e., overt, and physical aggression) [14]. In addition, another meta-analysis stated that violent video games increase aggression, in terms of thoughts, feelings, and behaviors. They found that exposure to violent video games is positively correlated with high levels of aggression among young adults and children, both male and female [23]. One proposed explanation of this finding is the social learning theory where humans are born with neurophysiological systems that allow them to act violently, but these mechanisms must be stimulated appropriately and are susceptible to cognitive control. For that, observational learning, rewarded performance, and structural variables contribute to acts of violence [24]. Thus, playing aggressive video games leads to the stimulation of aggressive behaviors because players imitate what they see. Furthermore, studies indicate a link between playing aggressive video games and players’ behaviors, resulting from players’ active involvement in a video game as opposed to watching violence on television where a viewer is not actively involved [25].

Considering gender differences, not surprisingly, we found that males play video games about twice as much as females and their aggression scores were significantly higher. This finding was already reported in the literature where young males spend more time and money than girls [26]. Additionally, boys enjoy action, sports, shooting, and strategy video games, while girls choose educational games. In terms of aggression, a study conducted in Japan found that playing violent video games increases hostility for males more than for females, with the reason hypothesized to be that males are more likely to be exposed to violent video games than females [27]. Another study also found that males, but not females, who reported playing video games excessively showed more aggressive behaviors [1]. The General Aggression Model asserts that aggressive behavior is dependent on distinct individual variables, such as gender, and environmental variables, such as the degree of violence in a video game, could provide an answer. As a child grows older and continues to be exposed to violent games or violent components of a game, changes in the chronic accessibility of aggressiveness-related knowledge and structures occur, which explains the increased aggression scores in older adolescents.

In Saudi Arabia, little is known regarding the use of video games by expatriate children. They are the sons and daughters of Saudi Arabia’s expatriate professionals, who make up 25% of the country’s current population [28]. Interestingly, non-Saudis make up almost 30% of our sample; however, we could not establish any statistical significance compared to Saudi nationals. Nonetheless, we found that non-Saudi students generally have higher mean verbal aggression scores, and those who played action and sports-type games reported even higher mean verbal aggression scores. Only one study tried to assess video game addiction in this specific population. Their study revealed that a considerable percentage (around 16%) were addicted to video games, and the correlation between video game addiction and psychological distress was robust [29].

The association between exposure to violent video games and aggression among youth remains debatable [13]. Despite that our study may focus on the aggressive behavior possibly correlated to video games, it is important to consider other explanations. Catharsis theory posits that playing video games can release aggressive behavior and have a calming effect by channeling latent aggression, and therefore video games can positively affect player behavior [30]. Multiple longitudinal studies concluded that exposure to violent video games is unrelated to youth aggression [13,31,32]. Furthermore, a recent meta-analysis showed that longitudinal studies do not appear to support long-term links between playing violent video games and youth aggression. Additionally, the correlations between them appear to be explained by methodological weaknesses and researcher expectancy effects [33]. In addition, two studies conducted by Graybill and his associates using a mixture of methodologies (self-reporting, experiments, and observation) claimed that video games can have short-term positive effects for children. These results are more consistent with catharsis theory and the idea that violent video games release aggressive impulses in an appropriate way [34,35]. Further studies focusing on assessing the baseline aggression for adolescents prior to and after using video games is warranted, to objectively assess the impact of video games on aggression and assess in depth gender differences.

## 5. Limitations

This study was limited by the lack of a control group and the lack of correlation with time spent playing. However, this was a preliminary study, and it was the first to be conducted in our region. It was also limiting by not addressing psychiatric illnesses, substance use, and physical illnesses as cofounding factors. Participants recall and observation biases are expected in this study design. The site of the study was limited to a metropolitan capital city; therefore, the results do not represent children across the country of different sociodemographic backgrounds.

## 6. Conclusions

Our results do suggest that playing video games has the effect of increasing adolescents’ aggressive behaviors, which is a conclusion that has also been supported by other studies.

Our recommendations are to educate parents on the pros and cons of playing video games, to raise awareness among adolescents on the negative impact of video games, and to encourage adolescents to play games that develop skills appropriate to their developmental levels and interests. Time spent playing video games should be scheduled and limited.

To conclude, more research needs to be conducted on the effects of playing video games on other significant issues, such as bullying, eating disorders, and cyber harassment. Moreover, other research could be conducted on the *positive* effects of playing video games, such as making friends, improving basic visual processes, and problem-solving skills.

## Figures and Tables

**Table 1 behavsci-12-00289-t001:** Demographic characteristics of participants (N = 485).

	Number (%) *
**Age (years)**	
Mean ± SD	16.5 ± 0.9
Sex	
Boys	234 (48.2)
Girls	251 (51.8)
Nationality	
Saudi	339 (69.9)
Non-Saudi	142 (29.3)
Caregivers	
Both biological parents	424 (87.4)
Single biological parent	36 (7.4)
Biological and step-parent	13 (2.7)
Relatives (other than parents)	9 (1.9)
Living arrangements	
Nuclear family (both parents, brothers and sisters)	441 (91.0)
Extended family (both parents, siblings, and other relatives)	31 (6.4)
How long you have been playing video games (years)	
Mean ± SD	7.4 ± 3.8
Currently playing video games	
Yes	345 (71.1)
No	139 (28.7)
Time spent playing video games per day (in hours)	
Mean ± SD	3.2 ± 3.5

* Percentages may not add up to 100 due to missing data.

**Table 2 behavsci-12-00289-t002:** Aggression scores of participants playing different types of video games.

Type of Video Game															
	Action Mean (sd.)			Adventure Mean (sd.)			Simulation Mean (sd.)			Sports Mean (sd.)			Strategy/Puzzle Mean(sd.)		
	Yes	No	*p*-Value	Yes	No	*p*-Value	Yes	No	*p*-Value	Yes	No	*p*-Value	Yes	No	*p*-Value
Anger	18.2 (5.2)	17.9 (5.6)	0.53	19.1 (4.9)	17.2 (5.6)	<0.001	19.2 (4.9)	17.8 (5.4)	0.02	17.2 (4.9)	18.4 (5.5)	0.01	19.1 (5.3)	17.6 (5.3)	0.005
Hostility	17.1 (5.8)	16.7 (6.1)	0.46	18.0 (5.8)	16.1 (5.9)	0.001	17.8 (5.7)	16.8 (6.0)	0.13	16.6 (5.9)	17.1 (5.9)	0.43	17.9 (5.6)	16.5 (6.0)	0.01
Verbal aggression	12.9 (3.6)	11.9 (3.8)	0.004	13.0 (3.5)	12.0 (3.8)	0.003	13.6 (3.6)	12.2 (3.7)	0.002	12.7 (3.7)	12.3 (3.7)	0.32	13.0 (3.5)	12.2 (3.7)	0.01
Physical aggression	19.6 (7.2)	17.5 (7.2)	0.002	19.9 (7.0)	17.9 (7.4)	0.009	19.3 (6.9)	18.6 (7.3)	0.44	20.0 (7.5)	18.2 (7.1)	0.01	18.9 (7.1)	18.6 (7.3)	0.75
Overall score	67.9 (17.4)	64.2 (18.7)	0.02	69.8 (16.5)	63.4 (18.8)	<0.001	70.0 (16.1)	65.5 (18.3)	0.03	66.6 (18.5)	66.2 (17.9)	0.41	69.0 (17.0)	65.0 (18.4)	0.02

**Table 3 behavsci-12-00289-t003:** Participant responses by age, sex, nationality, and types of video game played.

	Age			Sex			Nationality		
	15–16	17–18	*p*-Value	Male	Female	*p*-Value	Saudi	Non-Saudi	*p*-Value
**Participant responses to video game play time**									
How long you have been playing video games (years)	7.1 ± 3.6	7.5 ± 3.9	0.34	7.3 ± 3.4	7.4 ± 4.2	0.75	7.3 ± 3.7	7.4 ± 4.0	0.77
Video game play time per day (hours)	2.6 ± 2.2	3.6 ± 4.0	0.01	4.2 ± 4.0	2.2 ± 2.6	<0.001	3.4 ± 3.3	2.8 ± 3.9	0.25
Type of video game									
Action	131 (58.7)	138 (58.7)	1.00	152 (65.0)	127 (50.6)	0.002	200 (59.0)	76 (53.5)	0.31
Adventure	93 (41.7)	116 (49.4)	0.11	93 (39.7)	129 (51.4)	0.01	160 (47.2)	59 (41.5)	0.27
Simulation	43 (19.3)	44 (18.7)	0.90	38 (16.2)	50 (19.9)	0.34	61 (18.0)	26 (18.3)	1.00
Sports	59 (26.5)	74 (31.5)	0.25	122 (52.1)	19 (7.6)	<0.001	98 (28.9)	41 (28.9)	1.00
Strategy/puzzle	85 (38.1)	65 (27.7)	0.02	52 (22.2)	104 (41.4)	<0.001	98 (28.9)	56 (39.4)	0.02
How often do you play video games weekly									
Never	32 (15.0)	25 (11.7)	0.11	27 (11.8)	36 (16.0)	<0.001	39 (12.3)	24 (18.0)	0.01
1–3 times/week	93 (43.5)	75 (35)		75 (32.8)	103 (45.8)		116 (36.6)	61 (45.9)	
4–6 times/week	23 (10.7)	31 (14.5)		26 (11.4)	30 (13.3)		38 (12.0)	16 (12.0)	
Daily	66 (30.8)	83 (38.8)		101 (44.1)	56 (24.9)		124 (39.1)	32 (24.1)	
**Aggression scores according to age, sex, and nationality of participants**									
Anger	17.9 ± 5.4	18.3 ± 5.2	0.42	16.7 ± 5.4	19.4 ± 5.0	<0.001	17.9 ± 5.3	18.5 ± 5.4	0.24
Hostility	17.1 ± 6.2	16.9 ± 5.6	0.60	16.4 ± 6.3	17.5 ± 5.5	0.04	16.8 ± 6.1	17.4 ± 5.6	0.29
Verbal aggression	12.3 ± 3.7	12.7 ± 3.6	0.32	12.5 ± 4.0	12.4 ± 3.3	0.77	12.1 ± 3.7	13.2 ± 3.7	0.004
Physical aggression	18.1 ± 7.3	19.3 ± 7.1	0.04	20.1 ± 8.0	17.5 ± 6.3	<0.001	18.7 ± 7.3	18.7 ± 7.1	0.96
Overall score	65.6 ± 18.2	67.3 ± 17.5	0.30	65.7 ± 20.2	66.9 ± 15.8	0.49	65.6 ± 18.4	68.0 ± 17.5	0.19
**Aggression scores by type of video games played according to age, sex, and nationality**									
**Action**									
Anger	18.1 ± 5.2	18.3 ± 5.1	0.75	17.1 ± 5.2	19.5 ± 4.8	<0.001	18.0 ± 5.0	18.6 ± 5.6	0.39
Hostility	17.6 ± 6.2	16.6 ± 5.4	0.17	16.7 ± 6.1	17.6 ± 5.3	0.17	16.9 ± 5.8	17.8 ± 5.9	0.23
Verbal aggression	12.7 ± 3.4	13.0 ± 3.8	0.46	13.0 ± 3.7	12.8 ± 3.4	0.66	12.6 ± 3.5	13.6 ± 3.6	0.04
Physical aggression	18.9 ± 7.1	20.0 ± 7.2	0.20	21.1 ± 7.8	17.8 ± 5.8	<0.001	19.5 ± 7.2	19.8 ± 7.3	0.76
Overall score	67.4 ± 17.5	68.0 ± 17.6	0.76	68.0 ± 19.4	67.8 ± 14.8	0.92	67.2 ± 17.1	70.0 ± 18.5	0.23
**Adventure**									
Anger	19.3 ± 5.5	19.1 ± 4.3	0.77	17.5 ± 4.8	20.2 ± 4.7	<0.001	19.0 ± 4.9	19.3 ± 5.1	0.64
Hostility	18.8 ± 6.8	17.3 ± 5.0	0.07	17.0 ± 6.2	18.7 ± 5.5	0.03	17.8 ± 5.9	18.4 ± 5.9	0.52
Verbal aggression	12.7 ± 3.5	13.2 ± 3.5	0.26	13.3 ± 3.5	12.8 ± 3.5	0.36	13.0 ± 3.5	13.0 ± 3.3	0.93
Physical aggression	19.4 ± 6.9	19.8 ± 6.9	0.70	21.1 ± 7.7	18.6 ± 6.2	<0.001	19.8 ± 7.2	19.2 ± 6.6	0.54
Overall score	70.3 ± 17.7	69.5 ± 15.6	0.74	69.0 ± 18.4	70.4 ± 15.1	0.51	69.8 ± 16.8	70.0 ± 16.3	0.92
**Simulation**									
Anger	18.7 ± 5.5	19.8 ± 4.2	0.29	17.9 ± 4.3	20.2 ± 5.1	0.02	19.2 ± 5.1	19.5 ± 4.5	0.79
Hostility	18.1 ± 6.7	17.6 ± 4.5	0.72	18.1 ± 6.0	17.6 ± 5.5	0.69	18.4 ± 5.9	16.5 ± 5.0	0.14
Verbal aggression	13.7 ± 3.7	13.5 ± 3.6	0.71	15.5 ± 3.7	13.6 ± 3.6	0.89	13.4 ± 3.8	13.8 ± 3.0	0.65
Physical aggression	19.6 ± 7.3	18.6 ± 6.4	0.52	21.6 ± 7.8	17.4 ± 5.6	0.004	18.7 ± 7.3	18.7 ± 7.1	0.50
Overall score	70.2 ± 17.4	69.6 ± 15.0	0.87	71.3 ± 17.6	69.0 ± 14.8	0.51	70.1 ± 16.8	69.9 ± 14.9	0.96
**Sports**									
Anger	17.0 ± 5.5	17.3 ± 4.6	0.78	17.0 ± 4.9	18.3 ± 5.0	0.29	16.7 ± 5.0	18.2 ± 4.6	0.10
Hostility	17.2 ± 6.8	16.1 ± 5.2	0.28	16.5 ± 6.0	17.3 ± 5.1	0.57	16.1 ± 6.0	17.7 ± 5.6	0.15
Verbal aggression	12.0 ± 3.8	13.3 ± 3.5	0.04	12.8 ± 3.7	12.3 ± 3.6	0.58	12.4 ± 4.0	13.7 ± 2.7	0.02
Physical aggression	19.4 ± 8.0	20.4 ± 7.1	0.45	20.4 ± 7.7	17.6 ± 5.2	0.13	19.5 ± 8.0	21.3 ± 6.3	0.19
Overall score	65.8 ± 20.5	67.2 ± 17.3	0.66	66.8 ± 19.0	65.6 ± 15.7	0.79	64.8 ± 19.9	71.1 ± 14.5	0.04
**Strategy/Puzzle**									
Anger	18.5 ± 5.6	19.8 ± 4.8	0.13	18.1 ± 5.4	19.6 ± 5.1	0.11	19.1 ± 5.5	18.9 ± 5.1	0.82
Hostility	18.6 ± 6.1	17.2 ± 4.9	0.14	18.8 ± 6.7	17.8 ± 5.1	0.82	18.0 ± 5.9	17.7 ± 5.3	0.70
Verbal aggression	12.7 ± 3.5	13.5 ± 3.5	0.15	13.5 ± 3.5	12.8 ± 3.5	0.30	12.8 ± 3.4	13.4 ± 3.6	0.27
Physical aggression	18.0 ± 6.8	20.0 ± 7.3	0.08	22.6 ± 8.6	17.0 ± 5.4	<0.001	19.3 ± 7.3	18.2 ± 6.8	0.38
Overall score	67.8 ± 17.6	70.7 ± 16.6	0.31	72.3 ± 20.1	67.3 ± 15.1	0.08	69.3 ± 17.9	68.4 ± 15.8	0.74

**Table 4 behavsci-12-00289-t004:** Correlates of aggressive behaviors among Saudi youth.

	Anger		Physical Aggression		Verbal Aggression		Hostility	
	Odds Ratio (OR) ^¥^	95% CI	Odds Ratio (OR) ^¥^	95% CI	Odds Ratio (OR) ^¥^	95% CI	Odds Ratio (OR) ^¥^	95% CI
Action	1.9	0.6–5.4	3.5	1.1–20.1	2.4	0.5–11.1	3.8	0.6–11.6
Adventure	3.1	1.8–11.3	3.6	0.4–31.6	2.7	0.6–12.1	3.1	0.8–11.3
Simulation	3.1	0.4–24.4	2.6	0.6–18.7	2.9	0.8–21.1	3.2	0.4–24.9
Sports	2.2	0.6–7.7	0.8	0.1–4.6	1.1	0.2–5.6	1.1	0.3–3.4
Strategy/Puzzle	2.7	0.5–12.6	2.1	0.3–13.3	2.2	0.2–19.3	2.4	0.5–11.2

^¥^ Adjusted for age, gender, nationality, and duration of video games played.

## Data Availability

The data presented in this study are available upon reasonable request to the corresponding author. The data are not publicly available due to confidentiality reasons.

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
