# Peer review of "The Association between Video Game Type and Aggressive Behaviors in Saudi Youth: A Pilot Study"

_behavsci, 2022, doi:10.3390/bs12080289_

Round 1

Reviewer 1 Report

The topic is very interesting and there is a lot of recent bibligraphy

The abstract is confusing.The objective of the article is not stated. It should not include sections: method; results; conclusion.

 MINOR COMMENTS

1. Update the references

2. Change the abstract with the recommendations noted

Author Response

Update the references.

Thank you for your valuable feedback. The manuscript has been updated accordingly. 

Change the abstract with the recommendations noted.

Thank you for your feedback. The abstract has been updated accordingly. 

Reviewer 2 Report

Questions:

1.     Why did you include in the same paper both an APA citation procedure and what appears to be the numeric Vancouver style?

2.     With a large sample size of over 500, why would an emphasis on effect sizes be more informative than significance levels.

3.     Why did you cite a meta-analysis (reference #7) in support of the narrative that the APA considers violent video games as predictive of aggressive behavior? The article by Ferguson you referenced is unrelated to an APA position and moreover, the overall meta-analytic finding  in this article suggests that a media violence and aggression connection is unwarranted.

4.     Did you consider the Bonferroni procedure as a protection against type 1 errors when computing multiple statistical tests using the same dependent variable?

5.     Is a causal relationship tenable in a study with self-report data and no experimental manipulation?

Author Response

-Why did you include in the same paper both an APA citation procedure and what appears to be the numeric Vancouver style?

Thank you for your feedback. The references have been updated accordingly using a numeric citation as per MDPI referencing style. 

-With a large sample size of over 500, why would an emphasis on effect sizes be more informative than significance levels.

Thank you for your feedback. Effect size helps readers to understand the magnitude of differences found, a greater effect size indicates a larger difference between two groups. Statistical significance examines whether the findings are likely to be due to chance. We prefer to keep current reporting. 

-Why did you cite a meta-analysis (reference #7) in support of the narrative that the APA considers violent video games as predictive of aggressive behavior? The article by Ferguson you referenced is unrelated to an APA position and moreover, the overall meta-analytic finding  in this article suggests that a media violence and aggression connection is unwarranted.

Thank you for your feedback. We agree, and the citation has been corrected.

-Did you consider the Bonferroni procedure as a protection against type 1 errors when computing multiple statistical tests using the same dependent variable?

Thank you for your feedback. Yes, Bonferroni correction was conducted in order to protect from type 1 errors. 

-Is a causal relationship tenable in a study with self-report data and no experimental manipulation?

Thank you for your feedback. We agree, a casual relationship cannot be determined using a cross-sectional study with no experimental manipulation. This is not our intention as we are not aiming to establish causality. However, this is the first pilot study using this scale in Saudi Arabia. The second part of the study, which will include 5000 participants, is under production and will be submitted once we are done with the current study.

Reviewer 3 Report

The article is original as it deals with a very new, topical and very worrying issue for society.

The paper has a solid theoretical basis. But it uses few recent sources. Most of the references predate 2015. It is recommended to expand the introduction with more current citations between 2018-2022.

The work is very well structured, follows a methodology in line with the objectives set and is very complete. 

The results are clear and well represented. The conclusions respond to the planned objectives. 

I recommend expanding the introduction by providing more up-to-date information, including studies that defend the opposite hypothesis (video games do not influence aggressive behaviour). Add more references from 2018 to 2022.

I have also detected in the discussion some citations within the text that do not follow the form required by the journal.

Author Response

-I recommend expanding the introduction by providing more up-to-date information, including studies that defend the opposite hypothesis (video games do not influence aggressive behavior). Add more references from 2018 to 2022.

Thank you for your valuable feedback. The manuscript has been updated accordingly.  

-I have also detected in the discussion some citations within the text that do not follow the form required by the journal.

Thank you for your feedback. The references have been updated accordingly using a numeric citation.

Reviewer 4 Report

This pilot survey of Saudi youth in private and public schools in Riyadh, Saudi Arabia examined the associations between video game use, types of video game use and self-reported aggression. The study is well presented and apparently extends the literature on video gaming in the Middle East by examining these associations by including different types of video games and aggression. The following should be considered:

1. The authors should indicate in the title and introduction that the report is pilot and descriptive study only. 

2. More information should be provided about the 4 schools that refused to participate and did the principles provide reasons for refusing to participate. Are these schools that were already struggling with problems with violence?

3. The authors need to discuss how the video games were categorized into the various types and is the method of categorization reliable and reproducible? 

Author Response

-The authors should indicate in the title and introduction that the report is pilot and descriptive study only. 

Thank you for your feedback. The title has been updated accordingly. It reads now: “The Association between Video Game Type and Aggressive Behaviors in Saudi Youth: A Pilot Study”

-More information should be provided about the 4 schools that refused to participate and did the principles provide reasons for refusing to participate. Are these schools that were already struggling with problems with violence?

Thank you for your feedback. As it was a voluntary participation, no follow ups were pursued and the research team is not aware of violence problems in those schools.

- The authors need to discuss how the video games were categorized into the various types and is the method of categorization reliable and reproducible? 

Thank you for your feedback. It was based on the categorization of the General Authority for Audiovisual Media in Saudi Arabia. Accessible in https://ceservices.media.gov.sa/user/en/services/57

Round 2

Reviewer 2 Report

Among other concerns, two major issues were presented in the earlier review of the manuscript:

1.     Because of a large N, statistical analyses that attain statistical significance may be of trivial magnitude with little theoretical or practical import.  Hence, effect size estimates were recommended to supplement obtained significance levels. The response from the authors was, “we prefer to keep current reporting,” meaning significance levels alone would be reported, not effect sizes. Other than the stated preference, no other rationale was presented.

2.     Given a larger number of statistical tests, there is a need for the control of type one errors. In response to this concern, the authors responded that the Bonferroni procedure would be used in the revised manuscript. Yet, there is no evidence that that this assertion by the authors actually occurred. For example, even in the abstract itself, reference is made to three statistical tests for adolescents engaged in different types of video games on the same dependent variable of total self-reported aggression (lines 29,31, 34). Application of the Bonferroni procedure would indicate that a probability level of .02 is required for significance.  Nevertheless, the reported results were p<.05

More blatantly, on lines 160-161 the authors indicate that throughout the manuscript, p<.05 would be the significance level. This is contrary to the Bonferroni approach.

Conclusions: 

(A). Even though reliance on significance levels alone may disguise trivial findings, no modifications were made by fiat.

(B). The authors’ written statement was “Yes, Bonferroni correction was conducted in order to protect from type 1 errors.” There is no indication that this was indeed the case. 

Author Response

  1. Because of a large N, statistical analyses that attain statistical significance may be of trivial magnitude with little theoretical or practical import.  Hence, effect size estimates were recommended to supplement obtained significance levels. The response from the authors was, “we prefer to keep current reporting,” meaning significance levels alone would be reported, not effect sizes. Other than the stated preference, no other rationale was presented.

 Response:

Thank you for your feedback.

  • We have provided a statement about effect size (Cohen’s d) in ‘data analysis’ subsection (page 4, paragraph 1, line 7-8).   
  • We have included effect size statistics (Cohen’s d) in the text (result section) (page 4-6).      

  1. Given a larger number of statistical tests, there is a need for the control of type one errors. In response to this concern, the authors responded that the Bonferroni procedure would be used in the revised manuscript. Yet, there is no evidence that that this assertion by the authors actually occurred. For example, even in the abstract itself, reference is made to three statistical tests for adolescents engaged in different types of video games on the same dependent variable of total self-reported aggression (lines 29,31, 34). Application of the Bonferroni procedure would indicate that a probability level of .02 is required for significance.  Nevertheless, the reported results were p<.05

More blatantly, on lines 160-161 the authors indicate that throughout the manuscript, p<.05 would be the significance level. This is contrary to the Bonferroni approach.

 Response:

Thank you for your feedback.

  • We have provided a statement about Bonferroni correction in ‘data analysis’ subsection (page 4, paragraph 1, line 7).
  • According to Bonferroni correction –
  • We have revised the abstract (line 11-17).  
  • We have revised the result section of the manuscript (page 4-6).
  • We also adjusted part of discussion as a result.

Round 3

Reviewer 2 Report

My concerns about the manuscript have now been addressed appropriately.